# Microbiota and Oral Cancer as A Complex and Dynamic Microenvironment: A Narrative Review from Etiology to Prognosis

**DOI:** 10.3390/ijms23158323

**Published:** 2022-07-28

**Authors:** Pamela Pignatelli, Federica Maria Romei, Danilo Bondi, Michele Giuliani, Adriano Piattelli, Maria Cristina Curia

**Affiliations:** 1Department of Oral and Maxillofacial Sciences, Sapienza University of Rome, Via Caserta 6, 00161 Rome, Italy; 2Department of Medical, Oral and Biotechnological Sciences, “G. d’Annunzio” University of Chieti-Pescara, Via dei Vestini, 66100 Chieti, Italy; federicamaria.romei@gmail.com (F.M.R.); mariacristina.curia@unich.it (M.C.C.); 3Department of Neuroscience, Imaging and Clinical Sciences, “G. d’Annunzio” University of Chieti-Pescara, Via dei Vestini, 66100 Chieti, Italy; danilo.bondi@unich.it; 4Department of Clinical and Experimental Medicine, University of Foggia, Via Rovelli 50, 71122 Foggia, Italy; michele.giuliani@unifg.it; 5School of Dentistry, Saint Camillus International University for Health Sciences (Unicamillus), 00131 Rome, Italy; apiattelli51@gmail.com; 6Fondazione Villa Serena per la Ricerca, 65013 Città Sant’Angelo, Italy; 7Casa di Cura Villa Serena, 65013 Città Saint’Angelo, Italy

**Keywords:** oral cancer prognosis, oral microbiome, polymicrobial synergy, dysbiosis, interspecies communication, host-microbial interaction, oral squamous cell carcinoma

## Abstract

A complex balanced equilibrium of the bacterial ecosystems exists in the oral cavity that can be altered by tobacco smoking, psychological stressors, bad dietary habit, and chronic periodontitis. Oral dysbiosis can promote the onset and progression of oral squamous cell carcinoma (OSCC) through the release of toxins and bacterial metabolites, stimulating local and systemic inflammation, and altering the host immune response. During the process of carcinogenesis, the composition of the bacterial community changes qualitatively and quantitatively. Bacterial profiles are characterized by targeted sequencing of the 16S rRNA gene in tissue and saliva samples in patients with OSCC. *Capnocytophaga gingivalis*, *Prevotella melaninogenica*, *Streptococcus mitis*, *Fusobacterium periodonticum*, *Prevotella tannerae*, and *Prevotella intermedia* are the significantly increased bacteria in salivary samples. These have a potential diagnostic application to predict oral cancer through noninvasive salivary screenings. Oral lactic acid bacteria, which are commonly used as probiotic therapy against various disorders, are valuable adjuvants to improve the response to OSCC therapy.

## 1. Oral Microenvironment in Healthy Condition

The interspecies and host-microbial interactions can be included in the holistic framework of holobiont, which accounts for the host and its obligate or facultative symbionts, whose dynamic interconnectedness entails emergent interactions and outcomes [1]. Despite the fact that the notorious 10:1 ratio of bacterial/human cells has been revised to the estimate of about 1.3:1 in a “reference man”, with about 38 × 10^12^ total bacteria in the human body [2], the biological role of the microbiota has not scaled down. The number of bacteria in the mouth is less than 1% of the colon bacteria number; however, bacterial concentrations in the saliva and dental plaque are high (about 109 and 1011, bacteria per mL content, respectively) [2]. Comprehensive information about the bacterial species present in the oral cavity is publicly available (Expanded Human Oral Microbiome Database (EHOMD), n.d.). Multi-omics approaches are currently available to study the complexity of polymicrobial environments, interspecies interactions and the mechanisms underlying these relationships [3].

Changes in microbial biomass in the oral cavity depend on the interaction between different microbial species in the biofilm. The oral biofilm matures through interactions between early colonizing microorganisms with later colonizers through several mechanisms, including coagulation, metabolic exchange, small-molecule signal-mediated communication, and exchange of genetic material. The numerous bacteria in the oral cavity are not uniformly distributed over all surfaces but proliferate differently in ecological niches depending on their metabolism in a healthy oral cavity [4].

Multiple oral anatomical sites influence the ecology of these habitats and create microbial environmental differences (Figure 1). The most abundant species of buccal epithelium were *Streptococcus*, *Gemella*, *Eubacterium*, *Selenomonas*, *Veillonella*, *Actinomyces*, *Atopobium*, *Rothia*, *Neisseria*, *Eikenella*, *Campylobacter*, *Porphyromonas*, *Prevotella*, *Granulicatella*, *Capnocytophaga*, *Fusobacterium*, *Leptotrichia*, *Streptococcus mitis*, *Granulicatella adiacens*, often considered an opportunistic pathogen, was detected from bacteremia/septicemia in patients with infective endocarditis/atheroma [5] but also at buccal sites in the healthy oral cavity.

On the tongue dorsum between keratinized papillae filiformis there were species of *Streptococcus mitis*, *Streptococcus australis*, *Streptococcus parasanguinis*, *Streptococcus salivarius*, *Streptococcus* sp. clone FP015, and *Streptococcus* sp. clone FN051, *Granulicatella adiacens*, and *Veillonella* spp. Anatomical differences of the dorsum of the tongue from the lateral margin influenced the bacterial profiles above. The lateral margin of the tongue has a smooth nonkeratinized surface, and the predominant bacteria were *Streptococcus mitis*, *Streptococcus mitis* bv. 2, *Streptococcus* sp. clone DP009, *Streptococcus* sp. clone FN051, *Streptococcus australis*, *Granulicatella adiacens*, *Gemella haemolysans*, and *Veillonella* spp. [6,7]. The tongue species most associated with health were *Streptococcus salivarius*, *Rothia mucilaginosa* (*Stomatococcus mucilaginosus*), and an uncharacterized, cultivable species of *Eubacterium* (strain FTB41) [8].

Likewise, the main bacterial species were *Streptococcus mitis*, *Streptococcus mitis* biovar 2, *Streptococcus* sp. clone FN051, *Streptococcus infantis*, *Granulicatella elegans*, *G. hemolysans*, and *Neisseria subflava* on the hard palate. Streptococcus mitis was the most commonly found species in essentially all healthy sites and subjects. However, both *S. mitis* and *Streptococcus oralis* have been associated with bacterial endocarditis in patients with prosthetic valves [9,10]. In addition to the distinctive bacteria of a healthy oral cavity there were bacteria associated with periodontal disease, such as *Porphyromonas gingivalis*, *Tannerella forsythia*, and *Treponema denticola* [11].

**Figure 1 ijms-23-08323-f001:**
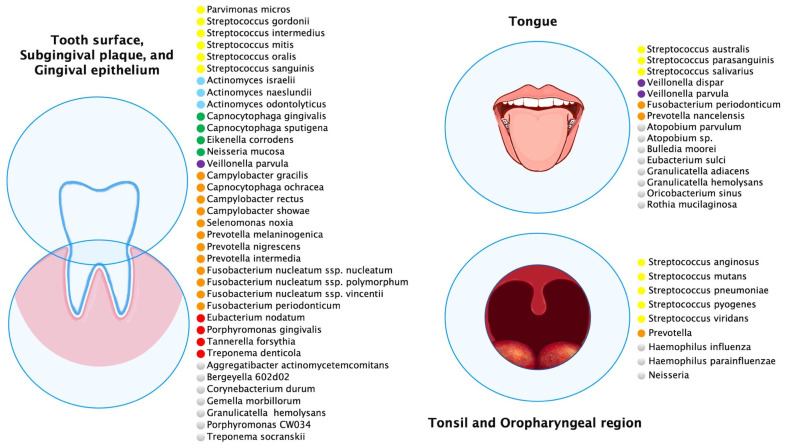
Predominant microbial communities within different ecological niches of the oral and oropharyngeal region. This figure was created using data sources from Aas et al. [6], Dewhirst et al. [12], Diaz et al. [13], Hall et al. [14], and Tamashiro et al. [15]. The species were color-coded according to microbial complexes described by Haffajee et al. [16].

## 2. Microbiota and Inflammation in The Oral Cavity

Oral health firstly depends on a correct balance between the various microbial populations that constitute the oral microbiota, when this balance is lost, a condition of dysbiosis is established. A condition of dysbiosis occurs, in fact, as a consequence of the disproportionate proliferation of some types of pathogenic microorganisms and it can have several triggers such as antimicrobial agents, age, hormonal changes, poor oral hygiene, smoking, or it may be favored by other body pathologies. Dysbiosis can lead to the generation of numerous oral diseases, including carious pathology, periodontal disease, and gingivitis [17].

Both periodontitis and gingivitis are characterized by inflammation, but periodontitis is a more serious condition than gingivitis. In the latter, the inflammation is limited to the soft tissues surrounding the teeth while periodontitis is a chronic disease which results in the destruction of teeth supporting structures and it is recognized as the leading cause of tooth loss in the adult population of industrialized countries.

Periodontal disease is defined as a chronic inflammatory disease and the bacteria responsible for this disease can persist, resistant to immune reactions, especially if the disease is not treated. Inflammation is part of the body’s first line of defense against invasive pathogens. A correct inflammatory response guarantees the correct resolution of the inflammation, but when the inflammatory reactions are inadequate and persistent over time there can be serious damage both locally and systemically.

Immune responses are subjected to a complex regulation that is the result of interactions between different types of immunocompetent cells, which communicate to each other through cytokines which have the function of regulating, suppressing, or amplifying the mechanisms of inflammation. Inevitably, periodontitis leads to an increase of proinflammatory cytokines such as interleukin- (IL-) 1α, IL-1 β, tumor necrosis factor-alpha (TNFα), and IL-6. IL-8 is a chemokine, important for the recruitment of defense cells in areas where their presence is necessary. IL-17 also seems to be involved in the pathogenesis of periodontal disease. It is a proinflammatory cytokine that stimulates the production of other mediators, such as IL-6 and matrix metalloproteinases (MMPs). The prostaglandins E2 (PGE2), which are vasoactive amines derived from arachidonic acid, are also involved in the inflammatory process. They act on fibroblasts and osteoclasts by stimulating the production of MMPs. The helper T cells regulate both humoral and cell-mediated immune responses. The humoral immune response is promoted by Th2 cells, which produce IL-4, IL-5, IL-10, and IL-13. Th1 lymphocytes release IL-2 and interferon-γ (INF-γ) which increases cell-mediated responses. The relationship between inflammation and cancer is not new. Recently, a very high number of studies have discussed this issue. Many malignancies arise from areas of infection and inflammation, simply as part of the normal host response. Indeed, there is a growing body of evidence that shows that many malignancies are initiated by infections; upwards of 15% of malignancies worldwide can be attributed to infections, a global total of 1.2 million cases per year [18,19]. P53 mutations are seen at frequencies similar to those in tumors in chronic inflammatory diseases such as rheumatoid arthritis and inflammatory bowel disease [18].

Undoubtedly, inflammation plays an important role in carcinogenesis and interestingly, several oral bacteria have been shown to activate inflammatory pathways associated with several stages of cellular transformation. Pathogenic bacteria can promote malignant tumors, and the tumor microenvironment itself can selectively stimulate the growth of bacteria [20]. The robust relationship between bacteria, chronic inflammation, and tumors is established: the functionally proinflammatory bacteriome in the body of the tumor is associated with oral cancer [21].

Antiapoptotic pathways—such as the JAK/STAT and phosphatidylinositol 3-kinase (PI3K)/Akt—can be activated by infected gingival epithelial cells (GECs). *Porphyromonas gingivalis*, a bacterium involved in periodontitis, can be internalized by GECs. Both pathways are also related to inflammation. Some cytokines such as IL-6, TNF-α, or IFN-γ function through the JAK/STAT pathway; additionally, the JAK/STAT pathway activates NF-κB and stimulates TNF-α production. The PI3K/Akt pathway, on the other hand, is involved in the increase of Toll-Like receptor-4 (TLR4) mRNA, in response to bacterial lipopolysaccharide (LPS). Finally, phosphorylation of Akt and its consequent activation induces NF-κB, which increases the transcription of antiapoptotic genes [19]. Coinfection studies using *Fusobacterium nucleatum* and *Porphyromonas gingivalis* show that they induce a synergic virulence response with a stronger inflammatory response triggered by elevated levels of TNF-α, NF-κB, and interleukin IL-1β, as well as higher levels of attachment and invasion into host cells [19].

Another interesting role is played by MMPs, which are a family of zinc-dependent endopeptidases with more than 20 different members. Matrix metalloproteinases play a crucial role in extracellular matrix and basement membrane degradation. This property favors periodontal tissue destruction, as well as cancer progression and metastasis, by causing tissue dissolution that enables tumor invasion [22]. Different studies have demonstrated higher concentrations of MMP-9 in patients with chronic periodontitis than in controls. MMP-9 is a widely investigated MMP, it is involved in cancer cell invasion, tumor metastasis, angiogenesis, and endothelial–mesenchymal-transition (EMT). It is also involved in mediating tumor microenvironment and modulating tumor-associated inflammation via cytokines and their receptors. The overexpression of MMP-9 has often been observed in different malignant tumors [23,24,25].

Another important role in the link between bacteria-inflammation and cancerogenesis is the production of free radical species. Different studies have shown that periodontitis is related to excessive reactive oxygen species (ROS) production or elevated oxidative damage [26].

Leukocytes and other phagocytic cells can induce DNA damage or interfere with DNA repair mechanisms, through their generation of reactive oxygen and nitrogen species that are produced normally by these cells to fight infection. These species react to the formation of peroxynitrite, a mutagenic agent. The generated products, in turn, demonstrate a strong affinity for more inflammatory cells, perpetuating the vicious cycle.

Many proinflammatory cytokines (such as IFN-γ, IL1β and other cytokines) facilitate induction of the inducible isoform of nitric oxide (NO) synthase (NOS), thus they could mediate excessive production of NO. NO facilitates vascular permeability, which accelerates nutritional supply to the tumor tissue and hence sustains the rapid tumor growth. This evidence suggests that inflammatory responses induced by various pathogens would accelerate mutagenesis as well as tissue damage [18,27]. Bacteria and host might interact according to the plaque ecological hypothesis [28]. Accumulation of supragingival and subgingival biofilms leads to inflammation, promoting alteration in physiological microbial composition. Such accumulation also increases the competitiveness of the oral pathogen at the expense of oral health-associated species through increased up-regulation of virulence factor expression. This results in a positive feedback loop.

The published studies on periodontitis and cancer mostly point to a positive association. In this association, inflammation has an essential role but more in-depth studies are needed. Sustained cell proliferation in an environment rich in inflammatory cells, activated stroma, and DNA-damage-promoting agents, certainly potentiates and/or promotes neoplastic risk. It is unclear whether the inflammatory mediators are critical for the development and growth of tumors or whether they constitute a permissive environment for the progression of malignancies. Moreover, a large number of studies show how the treatment of periodontitis substantially decreases markers of inflammation, but more investigation is also needed to assess how improved periodontal disease prevention and management strategies may impact cancer risk [18,22].

## 3. Microbiota and Pathogenetic Mechanisms Underlying Oral Squamous Cell Carcinoma (OSCC)

Head and neck cancers account for five percent of all tumors, and half of them occur specifically in the oral cavity [29]. Oral squamous cell carcinoma (OSCC) is a subset of head and neck squamous cell carcinoma, constituting over 90% of all oral cancers [30]. Despite advances in surgical techniques, adjuvant radiotherapy, and chemotherapy, the incidence of OSCC appears to be increasing worldwide, and the 5-year overall survival rate remains low, at approximately 50–60%. Smoking, drinking, and chewing betel are the main risk factors for oral cancer [31]. Other possible risk factors may include viral infection, fungal infection, and chronic periodontitis, whereas some cases cannot be clearly explained by any known risk factors [32,33,34]. Oral carcinogenesis is also associated with bacteria [35,36].

The correct balance between the commensal microbes and the host is essential for maintenance of physiological homeostasis, response to environmental changes, and survival. The composition of the microbiota at various anatomical sites is controlled by host genetics, particularly by the polymorphisms in immune-related genes, as well as by environmental factors, such as lifestyle and nutrition.

The microbiota has important local effects such as barrier fortification and the establishment of mucosal immunity, but it also exerts systemic effects, including modulation of metabolism inflammation and immunity. At the epithelial barrier surfaces, the composition of the microbiota and the abundance of particular species affects both inflammation and immunity, as well as the homeostasis of epithelial and stromal cells [37].

Human body surfaces are subject to constant environmental insult and injury. Infections, trauma, dietary factors, and germline mutations can contribute to breach of the body’s mucosal barriers. In most individuals, barrier breaches are rapidly repaired and tissue homeostasis is restored. Impaired host or microbial resiliency contributes to persistent barrier breach and a failure to restore homeostasis. In these settings, the microbiota may influence carcinogenesis by altering host cell proliferation and death, perturbing immune system function, and influencing metabolism within a host [38] (Figure 2).

For these reasons, it is so important to avoid a dysbiotic condition. Oral dysbiosis produces a microecological imbalance into the oral cavity, promoting a cascade of pathophysiological events which affect both evolution, development, progression, and metastasis of cancers [39].

Although cancer is generally considered to be a disease of host genetics and environmental factors, microorganisms are implicated in a consistent percentage of human malignancies. Infection-induced cancer accounts for approximately 16% of the global burden of all human cancers, corresponding to approximately 2 million new cases per year [37].

How microbes and the microbiota contribute to carcinogenesis, whether by enhancing or diminishing a host’s risk, fall into three broad categories: (1) altering the balance of host cell proliferation and death, (2) influencing immune system function, and (3) influencing metabolism of host-produced factors, ingested foodstuffs, and pharmaceuticals [38].

Different microbial agents are classified as human carcinogens. The herpesvirus Epstein-Barr virus (EBV) is associated with a diverse range of tumors, such as Burkitt lymphoma (BL), Hodgkin lymphoma (HL), post-transplant lymphoproliferative disorders (PTLD), diffuse large B cell lymphoma (DLBCL), NK/T cell lymphoma, nasopharyngeal carcinoma (NPC), and EBV-positive gastric cancer (EBV-GC) [40].

HBV and HCV are associated with hepatocellular carcinoma (HCC), they establish a chronic liver infection and in the 80% of cases they lead to HCCs. The role of the gut microbiota in regulating liver pathology and progression to HCC in mice has been clearly documented; interestingly, young mice, similarly to neonates or young children, fail to clear HBV infection in a hydrodynamic transfection model until an adult-like gut microbiota is established [37].

*Helicobacter pylori* is strongly associated with gastric carcinoma, although in most cases this infection leads to gastritis. Different strains of HPV (primarily HPV16 and HPV18) are also associated with anogenital cancers, a subset of head and neck cancers, and skin cancers. Interestingly, from several studies it has emerged that oncogenic viruses require inflammation to promote tumorigenesis [37,41]. Both HPV and EBV have been reported as oncovirus interplaying with local microbiome to promote host inflammatory carcinogenesis in head and neck cancers [42].

*Fusobacterium nucleatum* is a gram-negative anaerobic bacterium associated with various pathological conditions such as periodontitis, premature birth, inflammatory bowel diseases, and colon cancer. In several studies regarding the correlation between *Fusobacterium* and colic carcinogenesis, it has been observed how this bacterium influences the creation of a pro-inflammatory tumor environment. Growing evidence supports the possible causative role of *Fusobacterium nucleatum* in both cancer initiation, disease progression, and chemotherapy resistance [43]. All these correlations highlight the importance that inflammation and microorganisms have in carcinogenesis.

The mechanisms by which microbes influence cancer development and progression are microbic proliferation, bacterial toxins, β-catenin signaling alterations, and inflammation. Bacterial toxins can directly damage host DNA. Bacteria also damage DNA indirectly via host-produced reactive oxygen and nitrogen species. When DNA damage exceeds host cell repair capacity, cell death or cancer-enabling mutations occur. β-Catenin signaling alterations are a frequent target of cancer-associated microbes. β-catenin signaling results in activation of genes that control cell survival and proliferation. For example, *Fusobacterium nucleatum* is a member of the oral microbiota and it is associated with human colorectal cancer. Binding of the FadA adhesin of *Fusobacterium nucleatum* to E-cadherin activates β-catenin signaling, resulting in activation of genes that control cell survival and proliferation.

Proinflammatory pathways are engaged upon mucosal barrier breach in an evolving tumor. Loss of boundaries between host and microbe engages pattern recognition receptors and their signaling cascades. Feedforward loops of chronic inflammation mediated by NF-κB and STAT3 signaling fuel carcinogenesis within both transforming and non-neoplastic cells within the tumors [38]. It is important to know that the enrichment of a microbe at a tumor site does not connote that a microbe is directly associated with the disease. Some microbes can find favorable conditions for their survival in the tumor microenvironment.

Lifestyle risk factors trigger oral dysbiosis, whose inflammatory and genotoxic processes are again influenced by lifestyle risk factors [44]. External pressures trigger dysbiosis as a disturbance of the balanced equilibrium of the bacterial ecosystems in the human microbiome; with regards to this, tobacco smoking and chewing, psychological stressors, and diet all affect the oral microbiome and the onset and progression of periodontal diseases [45]. In this vein, poor oral hygiene, alcohol consumption, the use of betel quids, and genetic factors also influence the association between periodontopathogenic bacteria and oral cancer [46]. The smoker’s oral microbiota represented a significant abundance of *Veillonella dispar*, *Leptotrichia* spp., and *Prevotella pleuritidis* when compared to non-smokers. Heavy smokers had a greater abundance of *Fusobacterium massiliense*, which exhibited substantial sequence similarity with *Fusobacterium nucleatum* and *Prevotella bivia* [47]. Smoking may affect oral health by creating a different environment by altering connections among oral microbiota, and the microbiota and their metabolic function. Smoker-enriched bacteria can increase the acidity of the oral cavity promoting the release of amino acid-related enzymes, and amino sugar and nucleotide sugar metabolism [48].

All in all, lifestyle risk factors disturb the holobiont tuned equilibrium, triggering periodontal disease and dysbiosis into the oral cavity, which facilitate the onset and progression of oral cancer. Once these and/or other factors have led to the establishment of a tumor environment, the inflammation and hypoxic niche alter the oral microbiome determining a vicious loop. Alteration of the balance between bacteria and human hosts can increase the risk of oral cancer, even without taking into account environmental factors such as tobacco and alcohol.

Knowledge of the alterations in the oral microbial flora can help in the development of antimicrobial therapies useful for the prevention of OSCC. Overall oral bacterial profiles showed significant difference between cancer sites and normal tissue of OSCC patients, which might be considered diagnostic markers and treatment targets.

## 4. Bacterial Communities Associated with OSCC

Studies have shown that compared to healthy subjects, periodontitis-correlated taxa were significantly increased in the microbiota of the OSCC [49], including *Porphyromonas gingivalis*, *Fusobacterium nucleatum*, *Pseudomonas aeruginosa* [36], *Fusobacterium periodonticum*, *Aggregatibacter segnis*, *Campylobacter rectus* [50], *Campylobacter showae* [51], *Peptostreptococcus stomatis* [20,52], *Peptostreptococcus micros*, and *Catonella morbi* [20] (Figure 3a), while *Streptococcus*, *Veillonella*, and *Rothia* were significantly decreased in cancer tissue [53]. Loss of dental elements and the presence of disease have been associated with reduced richness and low diversity in the microbiome [54]. Most of the pathogenic periodontal bacteria are obligate or facultative anaerobes, and their abundance changed significantly in the hypoxic tumor microenvironment. In addition, the microbiota within the tumorous mucosa were saccharolytic and aciduric species. Host proteins may also be metabolized or fermented into sulfides and nitrosamines by *Firmicutes* and *Bacteroides*, potentiating cell mutations [55]. The microbiota composition appeared different depending on the type of sampling and depending on the stage of OSCC.

Within the OSCC tissue, *Solobacterium moorei*, hydrogen sulfide producer *Fusobacterium naviforme*, and *Neisseria flavescens* were significantly increased [56]. These bacteria can promote invasion across the basement membrane in OSCC; this is possible because (1) the volatile sulfur compounds produced can increase ROS release by inhibiting the enzyme superoxide dismutase and (2) methyl mercaptan promotes degradation of type 4 collagen [57]. *Fusobacterium periodonticum* could potentially cooperate with *Fusobacterium nucleatum* in tumor progression within the tumor tissue [49,58]. *Fusobacterium nucleatum* has the ability to coagulate with a diverse range of bacteria. In an aerobic environment it can coagulate with *Porphyromonas gingivalis* in cultures containing *Actinomyces oris* and *Veillonella* sp. In addition, *Fusobacterium nucleatum* helps generate a reducing and capnophilic environment that is necessary for the growth of *Porphyromonas gingivalis* [59]. The interaction between bacteria may or may not be favorable for the proliferation and survival of the species.

For example interaction of *Porphyromonas gingivalis* with *Streptococcus gordonii* may not be favorable because *Porphyromonas gingivalis* upregulates a series of genes involved in reduction of adhesion and signaling, and thus decrease its ability to form a biofilm [60]. There is also a negative correlation between the *Porphyromonas gingivalis* and *S. oralis*, *S. cristatus*, *S. intermedius*, or *S. mutans* because *Porphyromonas gingivalis* can reduce the production of extracellular arginine deiminase proteins by *Streptococcus cristatus* and *Staphylococcus intermedius* [61,62]. According to the “key pathogen” hypothesis [63], low-abundance oral bacteria may interact through multiple interspecies pathways, such as heterolactic fermentation, which may influence the growth of some bacteria and lead to dysbiosis of the oral microbiota. Dysbiosis of the microbiota promotes the development of OSCC. Bacteria contribute to the maintenance of homeostasis of the oral microenvironment.

There was a different composition of the microbiota of the tissue biopsy samples depending on the location of the tumor, for example *Capnocytophaga gingivalis*, *Rothia mucilaginosa*, and *P. intermedia* were significantly enriched in the lining mucosa, tongue, and gingiva, respectively [64]. These bacteria can secrete peptidases in tumor sites that are activated through proteinase-activated receptors (PARs) [65,66]. In this way, these protease-producing bacteria can degrade host tissue like extracellular matrix (ECM), destruct host physical barriers, and modulate host immune response, finally contributing to the onset and progression of tumors [67].

Microbioma analysis of tumor tissues versus normal buccal mucosa of OSCC patients using the 16S rDNA sequencing revealed an increase of genes related to cell motility in tumor sites, such as bacterial chemotaxis and flagellar assembly, genes associated with proinflammatory bacterial component, such as lipopolysaccharide biosynthesis, and genes involved in metabolism of cofactors and vitamins [20].

Intersample variation of OSCC oral microbiome was significantly associated with site of sampling, i.e., tumor site or buccal site far distant from the tumor site. in addition to the increase and decrease of oral bacterial species, some low-abundance oral bacteria might contribute to development of OSCC. A lower abundance of *Streptococcus* genera was observed in patients with OSCC, associated with an oral health condition. There were significantly higher abundance of *Streptococcus infantis* in smokeless tobacco non-consumers compared to that in smokeless tobacco consumers and contralateral buccal site of OSCC samples compared to that in the OSCC tumor site [68]. *Streptococcus sanguinis*, associated with periodontal health [69], could reduce the colonization of soft tissue surfaces by *A. actinomycetemcomitans* [70]. However, it is not clear whether *Streptococcus* genera actually promotes oral health or survives exclusively in healthy microenvironments [71]. *Streptococcus sanguinis*, despite promoting the adhesion of *Fusobacterium nucleatum*, reduces its production of H_2_O_2_ and its killing effect [72]. Other members of the oral microbiome that have been associated with a reduced risk of OSCC development included *Corynebacterium*, *Kingella*, *Leptotrichia*, *Neisseria*, *Parvimonas micra*, and *Haemophilus parainfluenza*. Their presence was suggested to be cancer protective [73]. However, in cancer, development metabolism and the functionally specialized role of the bacterial community are more relevant than composition [74].

While Capnocytophaga gingivalis, Prevotella melaninogenica, Sreptococcus mitis, Fusobacterium periodonticum, Prevotella tannerae, and Prevotella intermedia were significantly enriched in unstimulated saliva samples of OSCC patients (Figure 3b) [46,64,75,76,77,78,79]. Capnocytophaga gingivalis, Prevotella melaninogenica, and Streptococcus mitis are significantly elevated in the saliva of patients with OSCC with a diagnostic sensitivity and specificity of 80% and 82%, respectively [76].

*Neisseria* species, which are numerous in saliva samples, could play an important role in alcohol-related carcinogenesis because they produce acetaldehyde [80]. The bacteriome of saliva from patients with OSCC differed significantly from tumor tissue microbiota in terms of community structure, however, remained similar at taxonomic and metabolic levels except for elevated abundances of *Streptococcus*, *Lactobacillus*, and *Bacteroides*, and acetoin-biosynthesis, respectively [78]. Overabundance of *Porphyromonas gingivalis* in saliva was associated with advanced pathologic staging but lower recurrence rate of OSCC [81]. The origin of *Porphyromonas gingivalis* in OSCC tissue might be from the salivary microbial reservoir. Increased abundance of *Fusobacteria* species in oral tongue samples of OSCC patients was associated with significantly increased programmed death-ligand 1 (PD-L1) expression and—along with reduced abundance of *Rothia* and *Streptococcus* species—with lower alpha diversity [82].

**Figure 3 ijms-23-08323-f003:**
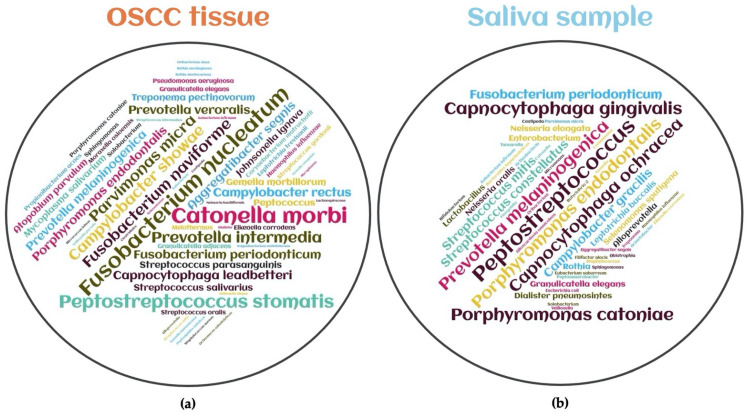
Word cloud of bacterial species detected in (**a**) cancer tissue and (**b**) saliva of OSCC patients. The PICo (Population, Intervention, Comparison and Outcomes) question was: “What are the bacteria significantly more abundant in OSCC patients when compared to healthy control?”. Articles were searched in Medline/PubMed, Science Direct, Google Scholar, or Scopus, with the string “oral microbiota” OR “oral bacteria” AND “oral squamous cell carcinoma” OR “OSCC”. Inclusion criteria were: in vivo studies, next generation sequencing (NGS) procedure, English language, date of publication: January 2005 to January 2022. Exclusion criteria were: animal studies, reviews and meta-analyses type. Data were finally retrieved from 13 articles [36,49,52,53,73,76,78,79,83,84,85,86,87]. Image created with WordArt (https://wordart.com/; last accessed on 26 June 2022). Bacteria found to be most frequently associated with OSCC are reported with larger font size; at progressively smaller font size, bacteria significantly associated with OSCC but less frequently reported.

## 5. Microbiome and Oral Cancer Prognosis

Multiple studies have revealed that the microbiota is predictive of disease severity and therapy outcome in cancers including but not limited to lung, pancreatic, colorectal, oral, breast, prostate, and liver cancers. In the last five years, several studies reported the relationship between *Fusobacterium nucleatum* and the clinical outcome in human cancers. For example, in the colon, differences in the flora of different intestinal segments may result in a different prognosis for colorectal cancer. *Fusobacterium nucleatum* is one of the potential pathogens associated with a worse prognosis and high *Fusobacterium nucleatum* levels were linked to a more advantage stage [88,89]. In terms of survival outcomes (disease-free survival and/or overall survival-OS-time) and in agreement with these data, other studies reported that adjuvant chemotherapy was more effective in patients with low *Fusobacterium nucleatum* levels than in those with higher *Fusobacterium nucleatum* levels [90,91]. Intriguingly, if the prognosis of colon tumors is different from the tumor site (left vs. right) [92] and different bacteria are found throughout the different segments of the intestine, this means that the bacteria influence the prognosis [93]. In regards to esophageal cancer, it was found [94] that *Fusobacterium nucleatum*–positive patients had significantly shorter cancer-specific survival and OS, and higher cancer-specific mortality compared with *Fusobacterium nucleatum*–negative cases. They also found an upregulation of the CCL20 chemokine and hypothesized that *Fusobacterium nucleatum* contributes to the acquisition of aggressive tumor behavior through chemokines activation. *Streptococcus anginosis*, in particular, has been associated much more with esophageal cancer rather than oral cancer [95]. Recently, it has been seen that even the breast tissue is affected by the microbiome. Dysbiosis may lead to the development of breast cancer through possible mechanisms linked to both immune regulation and the onset of a tumor microenvironment, but most of all to estrogen metabolism and the estrobolome [96], a term that indicates the collection of microbes that are capable of modulating the enterohepatic circulation of estrogens. The estrogen, conjugated in bile, can then be deconjugated by the enzyme β-glucuronidase secreted by gut bacteria. This active, unbound estrogen is reabsorbed into the circulation and binds to estrogen receptors which leads to several physiological responses, influencing reproductive processes and cardiovascular health. Variations or alterations in the microbiome cause higher circulating estrogen levels and increased risk of breast cancer [97]. Gut microbiome of breast cancer patients differs from that of healthy subjects and some species contain members that can cause double-stranded DNA breaks in cells that make up breast cancers [98,99]. Furthermore, a correlation has been shown [100] between composition and cancer grade, suggesting a relationship between the microbiome and breast cancer development and progression. Finally, balancing a dysbiotic microbiome with the use of antibiotics therapy, which usually increases the risk of breast cancer can be useful either before or after a cancer diagnosis [96].

As regards HNSCC, different bacteria including *Fusobacterium nucleatum* and *periodonticum*, *Streptococcus salivarius*, *Porphyromonas*, and different *Lactobacillus* subspecies are associated with the diagnosis of this kind of cancer. The periodontal pathogens *Fusobacterium nucleatum* (subspecies *polymorphum*), *Campylobacter subspecies*, *P. aeruginosa*, but also *Porphyromonas*, are considered as “mobile microbiome” because they originate in the OC but are also associated with extra-oral infections and inflammation. They have been reported to be associated with OSCC [36]. In particular, *Porphyromonas gingivalis*, *Prevotella intermedia*, *Fusobacterium nucleatum*, and *A actinomycetemconcomitans,* produce volatile sulfur compounds such as genotoxic and mutagenic agent hydrogen sulfide (H2S) in the gingival pockets that induce chronic inflammation and cells proliferation, migration, invasion, and tumor angiogenesis.

Bacteria such as *Fusobacterium nucleatum* and *Lactobacillus* play controversial roles in the development of oral cancer. The hypoxic nature of the tumor environment, the reduced immunity of the host, and the production of purines in the necrotic center of the tumor facilitate the attraction of bacteria to a tumor. As a consequence of hospitality, these bacteria disfavor tumor growth [101]. *Fusobacterium* is one of these bacteria. On the other hand, *Fusobacterium nucleatum* is implicated in the progression of OSCC by inducing proliferation of oral epithelial cell through activation of kinases and by binding its adhesion FadA to E-cadherin and in turn activating the Wnt/β-catenin pathway [102]. An interesting finding, although to be confirmed with further research, is that OSCC lesions associated with the abundance of *Fusobacterium*
*nucleatum* have a better prognosis when treated with drugs such as Pembrolizumab [103].

Lactic acid bacteria may have a protumoral and protective effect on OSCC. *Lactobacillus* has been implicated in OSCC development because they produce other acids, such as acetic, butyric, isobutyric, isovaleric, and isocaproic acids, which reduce the environmental pH, thus contributing to cancer growth and spread [104]. Furthermore, *Lactobacillus* was found to be increasingly abundant, relative to the increasing TNM staging of cancer patients [105].

Although lactic acid production can reduce the pH of the oral cavity by promoting cancer cell proliferation, on the other hand, they promote apoptosis (increases upregulation of PTEN and MAPK signaling), apoptosis provides antioxidative protection (against reactive oxygen species (ROS)) to cells, increased numbers of immune cells (T cells) and induction of cytokines (INF-gamma, TNF-alpha), and improved tumor suppression gene expression [106]. Lactic acid bacteria that can improve response to anticancer therapies are *Streptococcus mitis*, *Streptococcus gordonii*, *Streptomyces, Neisseria*, *Veillonella*, *Kingella*, and *Corynebacterium*. Lactic acid bacteria are commonly used in probiotic therapy and have shown success in the management of several disorders that include cancer [106].

*Lactobacillus fermentum* and *Lactobacillus plantarum* strains are the most studied bacteria as probiotic adjuvants in the treatment of OSCC. *Porphyromonas gingivalis* concentration can influence the prognosis of OSCC. *Porphyromonas gingivalis* overabundance in saliva has controversial prognostic power. According to some studies, patients with an overabundance of *Porphyromonas gingivalis* in saliva had a longer disease-free time and lower recurrence rate [81], while patients with a high *Porphyromonas gingivalis* level had the worst prognosis in esophageal squamous cell carcinoma [107,108]. The diversity of study populations and follow-up periods could have contributed to this prognostic inconsistency. The extent of inflammatory gene expression could also influence the prognosis of OSCC.

Among inflammatory genes, the COX-2 gene has been shown to be up-regulated in oral cancer and precancer, but its involvement in prognosis is uncertain. In the study by Abdolkarim Moazeni-Roodi et al., COX-2 induction was correlated with H-ras expression in OSCC, suggesting the involvement in oral cancer progression but also in increase of tumor grade [109]. In another study, a significantly higher expression of COX-2 was also found in oral cancer patients compared to normal controls, but COX-2 expression was found to be independent of grade of tumor and stage of disease [110]. The authors also suggest a role for COX-2 receptors in oral cancer carcinogenesis and provide for the use of COX-2 inhibitors for prevention of oral carcinogenesis.

The field of bacterial-based immunotherapies has recently received renewed interest, as they can infiltrate and replicate within solid tumors; in this vein, OSCC represents the ideal type of cancer to be treated [111]. Therapeutic strategies can be modeled targeting the virulence factors of *Porphyromonas gingivalis* and *Fusobacterium nucleatum* and developing phage therapy against oral pathogens such as *Fusobacterium nucleatum*, *Streptococcus mutans*, and *Neisseria meningitidis*. In addition, modulating pH microenvironment, along with taking advantage of probiotic bacteria such as *Streptococcus dentisani* or *Streptococcus A12* and of specific antimicrobial peptides are promising strategies to combat oral cancer as a multifactorial disease [112]. More attention should be paid on oral antibiotics, as they may alter intestinal microbiota, stimulating immune dysbiosis and limiting the abundance of probiotic bacteria, thus promoting OSCC development [113].

## 6. Limitations and Perspectives

The limitation of this review is a result of the difficulty of comparing studies with different methodologies. Indeed, the choice of analysis method and sample type greatly influence the results and do not allow us to perform a quantitative analysis.

Multicenter longitudinal association studies with third-generation sequencing platforms are needed to determine the etiopathogenic relationships between microbiota and OSCC. Therefore, we encourage further research in this up-and-coming field, which could lead to the development of effective prognostic targets and even therapies for patients with OSCC. Metagenomics and transcriptomics are indispensable for understanding the interaction between bacteria in the oral biofilm.

## 7. Conclusions

This review supports the association of oral microbiota with OSCC tumorigenesis and illustrates which bacteria could be used as biomarkers for the early diagnosis of OSCC. Microbial dysbiosis is evident in patients with OSCC. Several bacterial species appear to influence the progression, metastasis, and recurrence of oral cancer. The human microbiota may determine the response to cancer therapy through different mechanisms, and thus may positively or negatively influence the outcome. However, oral dysbiosis as a risk factor in the development and progression of oral cancer needs to be further elucidated. The genera, species, or combinations of bacteria involved have also not been thoroughly characterized.

This review provides insights into bacterial species or microbial compositions of potential importance to be investigated on saliva samples in patients with OSCC. Variation in individual species cannot discriminate between health and disease, so comparisons between complexes of microorganisms have now been included in the analysis. Microbial complexes significantly associated with oral carcinogenesis were depicted in an original way, rather than a summary list of oral bacteria to provide a direction for future clinical studies. Only studies performed through next generation sequencing (NGS) procedure were analyzed, as it requires a small volume of sample, eliminating the bias of PCR. The significantly increased bacteria in salivary samples were not the same as those identified within tissues of patients with OSCC. The use of the salivary microbiome as a reliable diagnostic tool has proven to be an important noninvasive opportunity in the early diagnosis of OSCC.

## Figures and Tables

**Figure 2 ijms-23-08323-f002:**
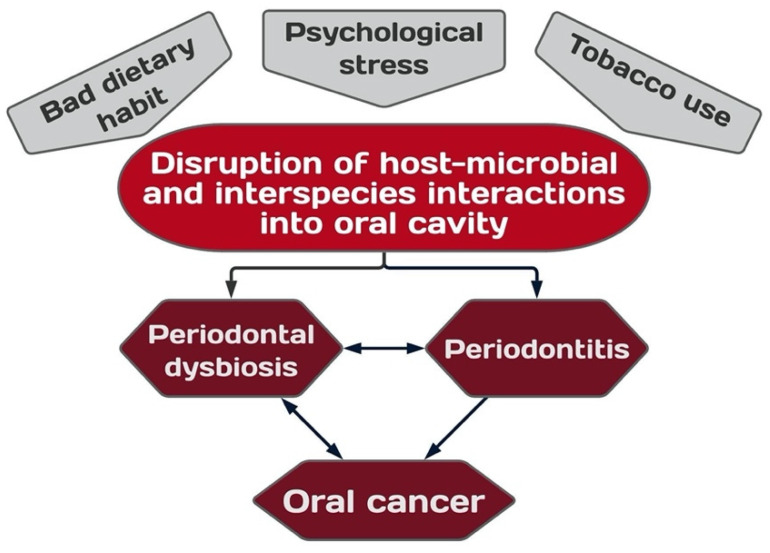
Pathological model of dysbiosis and oral cancer. Image created with BioRender (https://biorender.com; last accessed on 26 June 2022).

## Data Availability

Not applicable.

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
