# Peer review of "Microbiota and Oral Cancer as A Complex and Dynamic Microenvironment: A Narrative Review from Etiology to Prognosis"

_ijms, 2022, doi:10.3390/ijms23158323_

Round 1

Reviewer 1 Report

This manuscript presents very detailed aspects, from etiology to prognosis, regarding the microbiota and oral cancer, several bacteria being involved in influencing the progression, metastasis, and recurrence of oral cancer.

This review may help generate new ideas for the researchers in the evaluation of the response of cancer therapies, as long as microbiota may influence, by different mechanisms, in a positive or negative way the therapy.

This manuscript can be accepted for publication if the following aspects can be modified:

In all the text, Latin words should usually be printed in italics for clarity. Consider changing this aspect.

Line 25: These have potential diagnostic application to predict oral cancer

               These have a potential diagnostic application to predict oral cancer

Line 35: interconnectedness entail à entails

Line 36: bacterial / human …. Consider removing the extra spaces.

Line 37: into human body….. à into the human body

Line 41: Comprehensive information about the bacterial species present in the oral cavity are nowadays……. à ……..is nowadays

Line 52: metabolism in healthy oral cavity à metabolism in a healthy oral cavity

Line 68: [6,7].The tongue….. Consider adding a space before the punctuation.

Line 89: as: antimicrobial agents,… à remove the “:”

Line 91:  including: carious à remove the “:”

Line 95: chronic disease results à result

Line 108: tumor necrosis factor alpha à tumor necrosis factor-alpha

Line 130: The robust relationship between bacteria, chronic inflammation, and tumors is established, as the functionally proinflammatory bacteriome, despite different species composition, in the body of the tumor is associated with oral cancer [19]. Antiapoptotic pathways, such as the JAK/STAT and phosphatidylinositol 3-kinase (PI3K)/Akt, which inhibit the intrinsic pathway of apoptosis can be activated by infected gingival epithelial cells (GECs)…..

Hard to read text, please consider rephrasing the sentences.

Line 136: Both pathways have also been related à pay attention on the time

Line 146: MMPs that are a family à which are a family

Line 155: via cytokines and their receptors . à space before the punctuation to remove

Line 172: Accumulation of supragingival and subgingival biofilms leads to inflammation, promoting alteration in physiological microbial composition and increasing the competitiveness of the oral pathogen 174 at the expense of oral health-associated species through increased up-regulation of virulence factor expression.

Hard to read text, please consider rephrasing the sentences.

Line 178: In this association the inflammation has an essential à In this association, inflammation has an essential

Line 189: Head and neck cancers accounts for à Head and neck cancers account for

Line 224: The ways in which microbes and the microbiota contribute à How microbes…

This phrase may be wordy, consider changing the wording (concerning patients` survival..)

Line 230: has been found to be associated à is associated..

Line 268: In fact, some microbes can find favorable conditions for their survival in 268 the tumor microenvironment à Some microbes….

In Fact  may be unnecessary in this sentence. Consider removing it.

Line 270: Lifestyle risk factors trigger oral dysbiosis, whose inflammative and genotoxic processes are again influenced by lifestyle risk factors [39]. External pressures triggers dysbiosis as a disturbance of the balanced equilibrium of the bacterial ecosystems in the human microbiome; with regards to this, tobacco smoking and chewing, psychological stressors, and diet all affect oral microbiome and the onset and progression of periodontal diseases[40].

Hard to read text: inflammative à inflammatory; triggersà trigger; all affect the oral microbiome

Line 312: These bacteria can promote invasion across the basement membrane in OSCC because the volatile sulphur compounds produced can increase ROS release by inhibiting the enzyme superoxide dismutase and methyl mercaptan promotes degradation of type 4 collagen [52].

Hard to read text, please consider rephrasing the sentence.

Line 326:  is able to reduce à  Sounds better:  can reduce…

Pay attention to punctuation in all the text: for example, line 374: P.gingivalis

Conclusions need to be improved.

Overall, I suggest a minor revision of this work.

Author Response

This manuscript presents very detailed aspects, from etiology to prognosis, regarding the microbiota and oral cancer, several bacteria being involved in influencing the progression, metastasis, and recurrence of oral cancer.

This review may help generate new ideas for the researchers in the evaluation of the response of cancer therapies, as long as microbiota may influence, by different mechanisms, in a positive or negative way the therapy.

> Authors thank Reviewer 1 for his/her appreciation of the manuscript and his/her accurate suggestions to improve it. All comments have been addressed. Rev1 can find the revised parts tracked.

This manuscript can be accepted for publication if the following aspects can be modified:

In all the text, Latin words should usually be printed in italics for clarity. Consider changing this aspect.

Line 25: These have potential diagnostic application to predict oral cancer

               These have a potential diagnostic application to predict oral cancer

Line 35: interconnectedness entail  entails

Line 36: bacterial / human …. Consider removing the extra spaces.

Line 37: into human body…..  into the human body

Line 41: Comprehensive information about the bacterial species present in the oral cavity are nowadays…….  ……..is nowadays

Line 52: metabolism in healthy oral cavity  metabolism in a healthy oral cavity

Line 68: [6,7].The tongue….. Consider adding a space before the punctuation.

Line 89: as: antimicrobial agents,…  remove the “:”

Line 91:  including: carious  remove the “:”

Line 95: chronic disease results  result

Line 108: tumor necrosis factor alpha  tumor necrosis factor-alpha

Line 130: The robust relationship between bacteria, chronic inflammation, and tumors is established, as the functionally proinflammatory bacteriome, despite different species composition, in the body of the tumor is associated with oral cancer [19]. Antiapoptotic pathways, such as the JAK/STAT and phosphatidylinositol 3-kinase (PI3K)/Akt, which inhibit the intrinsic pathway of apoptosis can be activated by infected gingival epithelial cells (GECs)…..

Hard to read text, please consider rephrasing the sentences.

Line 136: Both pathways have also been related  pay attention on the time

Line 146: MMPs that are a family  which are a family

Line 155: via cytokines and their receptors .  space before the punctuation to remove

Line 172: Accumulation of supragingival and subgingival biofilms leads to inflammation, promoting alteration in physiological microbial composition and increasing the competitiveness of the oral pathogen 174 at the expense of oral health-associated species through increased up-regulation of virulence factor expression.

Hard to read text, please consider rephrasing the sentences.

Line 178: In this association the inflammation has an essential  In this association, inflammation has an essential

Line 189: Head and neck cancers accounts for  Head and neck cancers account for

Line 224: The ways in which microbes and the microbiota contribute  How microbes…

This phrase may be wordy, consider changing the wording (concerning patients` survival..)

Line 230: has been found to be associated  is associated..

Line 268: In fact, some microbes can find favorable conditions for their survival in 268 the tumor microenvironment  Some microbes….

In Fact  may be unnecessary in this sentence. Consider removing it.

Line 270: Lifestyle risk factors trigger oral dysbiosis, whose inflammative and genotoxic processes are again influenced by lifestyle risk factors [39]. External pressures triggers dysbiosis as a disturbance of the balanced equilibrium of the bacterial ecosystems in the human microbiome; with regards to this, tobacco smoking and chewing, psychological stressors, and diet all affect oral microbiome and the onset and progression of periodontal diseases[40].

Hard to read text: inflammative  inflammatory; triggers trigger; all affect the oral microbiome

Line 312: These bacteria can promote invasion across the basement membrane in OSCC because the volatile sulphur compounds produced can increase ROS release by inhibiting the enzyme superoxide dismutase and methyl mercaptan promotes degradation of type 4 collagen [52].

Hard to read text, please consider rephrasing the sentence.

Line 326:  is able to reduce   Sounds better:  can reduce…

Pay attention to punctuation in all the text: for example, line 374: P.gingivalis

> We have revised all these minor points as suggested

Conclusions need to be improved.

> Conclusion section has been improved

Overall, I suggest a minor revision of this work.

> Our manuscript has been improved thanks to the minor revisions suggested

Reviewer 2 Report

The review shows that there may be a possible association between dysbiosis and oral cancer,

This is an interesting review. However, there are some issues. The paper needs to be revised.

1) Please add words, “a narrative review” in the title following the guideline.

2) Please refer recent papers; doi: 10.3389/fimmu.2020.591088, doi: 10.1177/1533033819867354, doi: 10.1016/j.neo.2022.100813. Epub 2022 Jul 11, doi: 10.3390/cancers14133120, doi: 10.1016/j.molimm.2022.06.013, doi: 10.20517/cdr.2021.144. eCollection 2022, doi: 10.1016/j.micpath.2022.105638 and more. Then, the authors should clearly state what the paper add and the differences between this review and other reviews. It is unclear in the current form. It is very similar with one of the recent paper (doi: 10.1016/j.micpath.2022.105638) and the reviewer can’t find any new findings in this review.

3) Please change bacteria names to italic following the international rules.

4) Please revise the Figure 1 because the new data have been adding dramatically and evidence are updated compared to the papers at 2003-2012.

5) Please add the limitation of this review and also future directions.

Author Response

The review shows that there may be a possible association between dysbiosis and oral cancer,

This is an interesting review. However, there are some issues. The paper needs to be revised.

 > Authors thank Reviewer 1 for his/her time, interest and comments. We have addressed all the issues raised and we think the manuscript has been really improved. Rev1 can find the revised parts tracked.

1) Please add words, “a narrative review” in the title following the guideline.

> Added as sugggested

2) Please refer recent papers; doi: 10.3389/fimmu.2020.591088, doi: 10.1177/1533033819867354, doi: 10.1016/j.neo.2022.100813. Epub 2022 Jul 11, doi: 10.3390/cancers14133120, doi: 10.1016/j.molimm.2022.06.013, doi: 10.20517/cdr.2021.144. eCollection 2022, doi: 10.1016/j.micpath.2022.105638 and more. Then, the authors should clearly state what the paper add and the differences between this review and other reviews. It is unclear in the current form. It is very similar with one of the recent paper (doi: 10.1016/j.micpath.2022.105638) and the reviewer can’t find any new findings in this review.

> All the suggested papers have been evaluated and included (Rev2 can find them highlightd in yellow into the References section). Into the conclusions, we have stated what our manuscript add and how differ from similar works: we focused on manuscripts reporting the use of third-generation sequencing, also known as high throughput sequencing. In addition, we think our visual representation of results (i.e., the figure 3) is graphically original and it provides a more easy and appreciable interpretation of data.

3) Please change bacteria names to italic following the international rules.

> All bacteria names have been changed to italic style

4) Please revise the Figure 1 because the new data have been adding dramatically and evidence are updated compared to the papers at 2003-2012.

> Figure 1 has been updated - with novel bacteria included - and revised as suggested, removing the reference of Kazor et al. (2003) and adding the more recent ones of Hall et al. (2017), and Tamashiro et al. (2021)

5) Please add the limitation of this review and also future directions.

> Limitations and perspectives have been added in Section 6 (Conclusions)

Reviewer 3 Report

Authors Pignatelli et. al. have presented an interesting review in which they have discussed association between microbial community in oral cavity with the possibility of a development of oral cancer in the future. Through the article Microbiota and oral cancer as a complex and dynamic microenvironment: from etiology to prognosis, authors have managed to generate interest and attention toward a possible positive association between the microbiota association of oral cancer. They have discussed different factors that could play an important role in driving this process. The manuscript can be accepted.

Author Response

Authors Pignatelli et. al. have presented an interesting review in which they have discussed association between microbial community in oral cavity with the possibility of a development of oral cancer in the future. Through the article Microbiota and oral cancer as a complex and dynamic microenvironment: from etiology to prognosis, authors have managed to generate interest and attention toward a possible positive association between the microbiota association of oral cancer. They have discussed different factors that could play an important role in driving this process. The manuscript can be accepted.

> Authors thank Rev3 for his/her time and interest. We are really grateful of this favourable comment

Round 2

Reviewer 2 Report

The paper was overall improved. However, there is a minor point.

The authors should move the last paragraphs (L516-537) before the conclusion because one is a limitation paragraph and another is a future direction one, but not conclusion. Please add two paragraphs.

Author Response

The authors thank Rev2 for his/her appreciation of our revisions and once again for his/her time and helpful comments. In order to solve the minor issue raised, we have moved the sentences on limitations and perspectives from the part he/she mentioned to a new section before the Conclusions. The section 6 now contains these sentences in two paragraphs.